# Multimodal impact of copper-silicon hybrid nanotools towards bacterial leaf streak, wheat biochemistry and productivity parameters

Waqas Ahmad[1], Muhammad Atiq[1]*, Nasir Ahmed Rajput[1], Rizwana Maqbool[2], Jamil Shafi[3], Abdul Jabbar[4], Sohail Asad[5], Muhammad Jahanzaib Matloob[1], Hassan Zia[1], Muhammad Usama[1]

1 Department of Plant Pathology, Faculty of Agriculture, University of Agriculture, Faisalabad, Pakistan,
2 Department of Plant Breeding and Genetics, Faculty of Agriculture, University of Agriculture, Faisalabad, Pakistan, 3 Department of Plant Pathology, Constituent College Depalpur, Okara, University of Agriculture, Faisalabad, Pakistan, 4 Department of Agronomy, Faculty of Agriculture, University of Agriculture, Faisalabad, Pakistan, 5 School of Tea and Coffee, Pu'er University, Pu'er, Yunnan, P.R. China

* muhammad.atiq@uaf.edu.pk

## Abstract

Wheat production is substantially harmed by biotic and abiotic stress. Among biotic stresses, bacterial leaf streak (BLS) of wheat caused by bacterium *Xanthomonas translucens* pv. *undulosa* (*Xtu*) induces crop yield losses up to 10–40%. This multi-step investigation encompassed the evaluation of the antibacterial potency of parthenium-mediated copper nanoparticles (CuNPs) and silicon nanoparticles (SiNPs). Green synthesized Cu-Si nanoparticles were evaluated under lab and greenhouse conditions employing a Complete Randomized Design (CRD) and under field conditions adopting Randomized Complete Block Design (RCBD) alone and in combination. The scanning electron microscopy and X-ray diffraction showed that CuNPs and SiNPs exhibited semi-spherical and spherical morphology with average size of 61.49nm and 14.36nm, respectively. Among the tested nanoparticles, maximal inhibition one was expressed by CuNPs+SiNPs (19.06mm), followed by CuNPs (14.14mm) and SiNPs (10.56mm) as compared to control. The least disease incidence under greenhouse (14.75%) and field-scale evaluation (29.46%) was expressed by combined treatment CuNPs+SiNPs, followed by single treatment CuNPs and SiNPs in comparison with control. Moreover, the execution of the most effective dosage of CuNPs+SiNPs enhanced the amounts of enzymatic and non-enzymatic antioxidants like SOD, POD, CAT, $H_2O_2$, TPC, TSS by 3.09, 3.01, 1.87, 7.35, 3.61 and 1.08 µg/g, respectively. Similarly, most effective dosage of CuNPs+SiNPs increased the yield-related attributes of the wheat plant such as root length (4.52cm), shoot length (4.063cm), chlorophyll contents (5.56 SPAD), spike length (15.98cm), spikelets per spike (19.88) and Number of grains per spike (31.97), 100 grains weight (17.07g), plant height (81.46cm), stomatal conductance (288.69

**Data availability statement:** All relevant data are within the manuscript and its Supporting information files.

**Funding:** The author(s) received no specific funding for this work.

**Competing interests:** The authors have declared that no competing interests exist.

m.mol m$^{-2}$s$^{-1}$), and root biomass (492.15 kg/ha). Recent findings emphasize the potential of ecological nanotechnology-based tactics in plant disease management. Furthermore, nanoparticles applications with CuNPs and SiNPs were an eco-friendly tactic for managing the bacterial leaf streak of wheat and enhancing the antioxidant defense system and yield-related attributes of wheat.

## Introduction

Pakistan ranks eighth in production, yielding 31.4 million tonnes over an area of 9.4 million hectares. Wheat contributes 9.0% to agricultural value-added and 2.2% to GDP of Pakistan [1]. It is crucial for diets, furnishing key nutrients such as proteins, carbohydrates, vitamins, fats, minerals and nutrients [2,3]. It aids in treating heart diseases, constipation, cancer, obesity, and diabetes [2]. Wheat by-products (wheat bran, white flour, bread flour, cake flour, gluten flour, semolina, farina) are used in animal feed, bioethanol, baking, cosmetics, and pharmaceuticals [4].

Different biotic (fungi, bacteria, nematode, virus) and abiotic (drought, temperature, salinity, floods, radiations, pollutants) factors are responsible for the reduced production impacting qunatity and quality of crop [5]. Among the biotic factors, bacterial leaf streak (BLS) which is caused by the gram-negative *Xanthomonas translucens* pv. *undulosa* is a major global threat to wheat production [6,7]. Optimal growth was noted at 28–30°C, but it terminates at 36°C, particularly when moisture levels are high [8]. The disease can cause up to 10–40% yield losses worldwide, dependent on inoculum levels and host susceptibility [9]. The pathogen also endures on crop debris, soil, and weeds, penetrating the plants through stomata, wounds and hydathodes. Inside the plant, the pathogen multiply in the mesophyll tissues producing exopolysaccharides which lead to biofilm formation [10]. *Xtu* uses type III effectors (T3E) via a type III secretion system (T3SS) to initiate the disease symptoms [11] which appeared as water-soaked lesions that gradually turn yellow, brown, and necrotic. These lesions eventually ooze a honey-like exudate, which spreads under humid conditions and forms a thin, transparent layer [12].

Different management approaches are in practice to cope the bacterial leaf streak of wheat. The most efficient approach for managing the disease is the employment of resistant varieties. However, under favorable conditions, even these varieties become susceptible [13]. Excessive use of chemicals for disease management can disrupt plant functions and cause toxicity, leading to a range of health issues and organ impairments [14]. Considering the hazardous effects of chemicals and the limitations of other control measures, advanced strategies such as green nanotechnology are essential for effective disease management [15]. This emerging technique uses nanoparticles to treat plant bacterial disease being preferred over chemical and physical methods due to its cost-effectiveness and environmental friendliness [16,17]. Nanoparticles (1–100 nm) are used in small doses to effectively inhibit microbial enzymes, damage essential molecules, cause cell wall damage, and stimulate oxidative stress which led to microbial death [18,19]. Use of weeds for the

integration of green-based nanoparticles offer an eco-friendly approach to manage plant diseases [20]. Researchers have developed green-based nanoparticles, using weed extracts to control bacterial plant diseases [21]. Copper nanoparticles (Cu-NPs) demonstrated strong antibacterial activity against *Xanthomonas campestris* [20], and *Xanthomonas campestris* pv. *vesicatoria* [22]. Copper and silver nanoparticles synthesized from *Eucalyptus globulus* leaves were effective against *Xanthomonas citri* [13], and Cu-NPs also inhibited *Ralstonia solanacearum* [23]. Similarly, silicon nanoparticles (Si-NPs) proved effective against *Pectobacterium carotovorum* and *X. campestris* pv. *carotae* in carrots [24,25].

Nanoparticles (NPs) improve the wheat biochemistry by enhancing the nutrient acquisition, photosynthetic efficacy, and inducing stress resistance [26]. NPs regulate the enzymatic activity of antioxidants activity such as SOD, POD, and CAT that scavenge the ROS and combat the oxidative damage [27]. NPs also regulate the non-enzymatic antioxidants acitvity like $H_2O_2$, TPC, and TSS that make sure the strengthening of defense responses and reducing oxidative stress, and in result, ensuring the better overall plant health and resilience [28]. NPs not only suppress the harmful effects of pathogens but also promote the plant growth metrics [29]. So, the current research effort was directed to investigate the antibacterial function of *Parthenium hysterophorus* mediated Cu-Si hybrid nanotools towards bacterial leaf streak of wheat and its effect on the physiology and biochemistry of wheat plant.

## Materials and methods

### Synthesis of weed-based nanoparticles (NPs)

Parthenium (*Parthenium hysterophorus* L.) was collected from the Agronomy Department Research Area, and was shade-dried for about 4−5 days. After this, it was sun-dried over 3 days. Then, it was kept in a dry oven (101−1AB) to remove remaining moisture at 65°C for 4 hours. After oven drying, pestle and mortar was employed to grind the weed into fine powder. Then, powder was filtered by using muslin cloth to obtain a fine powder without impurities. 20 g of this fine powder was added in a beaker (250 mL) containing 100 mL methanol, covered with an aluminium foil and kept under darkness over a period of 24 hours (for effective extraction, deterrence of photodegradation and facilitates the uniform interaction). After this, the solution was stirred in a magnetic stirrer at 70°C for about 15 minutes (for uniform dissolution and helped in boosting the reaction). Then, filtration of the solution was done through Whatman's Filter Paper No. 41. For the preparation of Silicon and Copper nanoparticles, 17g sodium metasilicate ($Na_2SiO_3$) and 17g copper sulphate ($CuSO_4$) was added in the filtrate separately and put it on magnetic stirrer (C-MAG HS 4) for mixing of solution and then placed it in ultrasonic cleaner (VEVOR) at 60°C for breaking the bond for 1 hour. To obtain Si and Cu, nanoparticles in powder form, the solution was placed in a furnace oven (Carbolite PF60, higher temp. processing, uniform heating under controlled conditions) [13]. For the preparation of hybrid nanoparticles, both silicon and copper nanoparticles were mixed.

### Characterization of parthenium mediated nanoparticles

The characterization of the parthenium mediated nanoparticles was performed by adopting different techniques as performed by [30].

**UV–visible spectroscopy.** The biological reduction of the solution was pursued by determining its absorbance with a UV-visible spectrophotometer (V-750) over the range of 200–800 nm. 1 mL of the sample was used and 1mL of distilled water was used as control.

**X-ray diffraction spectroscopy.** The average size of nanoparticles was determined by employing diffractometer (Philips X-Pert MMP). The P Rofit software was employed to control the diffractometer on a computer, with molybdenum radiation (MoK) used as the source ($\lambda = 0.7093$ A). The Mo-tube operated at 50 kV and 40 mA to facilitate the bombardment process.

**Scanning electron microscopy.** The morphological characteristics of the nanoparticles synthesized through parthenium extract was assessed by employing scanning electron microscopy (SEM). For SEM analysis, A 1 mg sample

was deposited onto carbon-coated grids and imaged at different magnifications for the assessment of its morphological characters.

### *In-vitro* assessment of green synthesized NPs against *Xanthomonas translucens* pv. *undulosa*

The inhibition zone technique was used for the assessment of green synthesized nanoparticles (copper and silicon) from weed (*Parthenium hysterophorus* L.) against *Xanthomonas translucens* pv. *undulosa* under laboratory conditions. For this purpose, one litre bacterial growth media (NA) was prepared in the Laboratory. To get three concentrations (0.025, 0.05, and 0.075%) of NPs, 0.025, 0.05, and 0.075 g of (Cu and Si) powder alone and in combination (Cu + Si) were added in bottle (250 mL) containing 100 mL distilled water separately. After that, NA medium was poured into 9 cm Petri plates and allowed to solidify. Through autoclaved cotton swab, bacterial culture was streaked on Petri plates, and this whole procedure was performed within a laminar flow cabinet (RTVL1312, UK) to prevent the contamination. Circular pieces of the filter paper (1 cm) were cut and sterilized in an autoclave (Robus Technologies, RTA85) at pressure of 15 PSI and 121°C for 15 minutes. Then, these pieces of filter paper were dipped into formulated concentrations (0.025, 0.05, and 0.075%) of NPs (Cu and Si) alone and in combination. These dipped filter paper pieces were placed in the centre of the Petri plates having a culture of *Xanthomonas translucens* pv. *undulosa*. In case of control treatment, filter paper pieces were dipped in distilled water rather than in nanoparticle formulations. The plates were wrapped with paraffin and incubated (Heraeus) at 26 ± 2°C for optimal growth of bacterium.

The inhibition zone was measured by employing digital vernier calliper (500−196, Mitutoyo) after 24, 48, and 72 hours [13]. This trial was designed under Complete Randomized Design (CRD) adopting three replications of each treatment.

### *In-vivo* evaluation of green synthesized NPs against bacterial leaf streak of wheat

A field experiment was conducted following the methodology of [13] with trivial adjustments at the Department Research Area. PBG Line 3 was sown employing Randomized Complete Block Design (RCBD) by upholding the P-P and R-R spacing at 10–25 cm and 10–30 cm, respectively for better growth and development. Four treatments (CuNPs + SiNPs, CuNPs, SiNPs, and Control) with three replications of each treatment were evaluated against bacterial leaf streak of wheat. Aqueous suspension of the bacterium *Xtu* was prepared from culture grown for 48 hours and employing spectrophotometer (Hitachi U-2001, 121003), bacterial concentration @ $1 \times 10^8$ CFU/mL was determined.

The bacterium *Xtu* was artificially introduced in wheat plants by using spray method at the stage of 4–5 leaves. This was done early in the morning when maximum number of stomata were opened in the daytime. After the typical symptom's expression, most effective concentration (0.75%) of nanoparticles was sprayed in the field while, in the case of control, only distilled water was sprayed on plants. Data regarding the incidence of the bacterial leaf streak was recorded after 7, 14, and 21 days using the formula given below [31],

$$\text{Disease incidence} \ (\%) \ = \ \frac{No. \ of \ Diseased \ Plants}{Total \ No. \ of \ Plants \ Observed} \ x \ 100$$

### Alterations in the biochemical profiling of wheat leaves after application of nanoparticles

Both treated and untreated wheat plants were collected from the field, and their leaves were cut into small pieces. A 0.5 g leaf sample was ground by employing a sterilized pestle and mortar with potassium phosphate buffer (pH 5). Then, the mixture was centrifuged (TGL-16A) for 5 minutes at 12000 revolutions per minute (rpm). The resulting supernatant was used for advanced biochemical analysis.

**Determination of SOD (Superoxide dismutase) from treated and untreated wheat leaves.** For this purpose, a reaction mixture containing 200 μL methionine, 100 μL enzyme extract, 200 μL triton X, 500 μL potassium phosphate

buffer (pH 5), and 100 µL NBT (as an indicator for superoxide radicals $O_2^{\cdot-}$) with 800 µL distilled water was prepared. It was kept under UV light for 15 minutes. Absorbance was recorded after the addition of 100 µL riboflavin employing spectrophotometer (BioTek, 800TS) at 560 nm wavelength [32].

**Determination of POD (Peroxidase) from treated and untreated wheat leaves.** For the determination of POD from treated and untreated wheat leaves, a reaction mixture consisting of 20 mM guaiacol, 40 mM $H_2O_2$ (100 µL), 100 µL enzyme extract, and 800 µL potassium phosphate buffer (pH 5) was prepared. Absorbance was recorded employing spectrophotometer (BioTek, 800TS) at frequency of 470 nm.

**Determination of CAT (Catalase) from treated and untreated wheat leaves.** A reaction mixture (100 µL enzyme extract and 100 µL of 5.9 mM $H_2O_2$) was prepared to determine the concentration of catalase in wheat leaves. A spectrophotometer (BioTek, 800TS) was employed to record the absorbance at 240 nm.

**Determination of $H_2O_2$ (Hydrogen peroxide) from treated and untreated wheat leaves.** Leaves of treated and untreated plants were collected and sliced into small-sized pieces. A 50 mg leaf sample was taken along with trichloro acetic acid buffer (enhance extraction efficiency, stabilize and prevent degradation of $H_2O_2$), grinded, and then centrifuged (TGL-16A) at 4°C for 15 minutes at 12000 rpm (stability of sample and preservation of enzymatic content). Then the solution was filtered and treated with potassium phosphate buffer (pH 7) followed by potassium iodide buffer. The final mixture was placed in a digital incubator (RTI-250) for 5 minutes providing a controlled environment for optimal reaction conditions and by using microplate reader (BioTek, 800TS), absorbance was recorded at frequency of 390 nm.

**Determination of TPC (Total phenolic contents) from treated and untreated wheat leaves.** 100 µL enzyme extract was prepared to determine the TPC from treated and untreated wheat leaves. A reaction mixture was prepared consisting of 800 µL $Na_2CO_3$ (700 mM) and 200 µL of 10% Folin-Ciocalteu (F-C) reagent and allowed to react for 1 hour. Using microplate reader (Biotek, 800TS), reading was recorded at frequency of 765 nm.

**Determination of TSS (Total soluble sugars) from treated and untreated wheat leaves.** The 100 mg sample was subjected to hydrolysis through 5 mL of 2.5 N hydrochloric acid for 3 hours in water bath (HT, XMTD-204), then neutralized with sodium carbonate ($Na_2CO_3$) until bubbling ceased, and finally cooled at at room temperature. A solution of 100 mL was prepared and processed for centrifugation (TGL-16A). 0.5 µL resultant supernatant along with aliquots @ 1 mL was obtained and used for more studies. Special values were developed by using 0.20, 0.4, 0.6, 0.8, and 1 mL as operational values and 0 represented as a blank. To make the volume more than one mL, distilled water was used, and anthrone reagent @ 4 mL was added. For heating the solution, a water bath (HT, XMTD-204) which allows the consistent and controlled heating during hydrolysis without evaporation was used, and absorbance was measured employing spectrophotometer (BioTek, 800TS) at 630 nm [33].

## Assessment of impact of NPs on the agronomic attributes of wheat

Root length, shoot length, spike length and overall plant height were measured by manual method using measuring tape (HV-1455) in cm. SPAD (SPAD-502 Plus) meter was used for the precise determination of chlorophyll contents in the leaves of wheat plant by clamping it gently on the leaf. By detaching the spike from the stem, spikelets were counted manually. Total no. of grains per spike were counted precisely by manual method after the threshing process. For the determination of 100 grains weight, 100 grains were counted and then weighed on a weighing balance (SF-400A) in grams. The porometer (AP4 Porometer) was used to determine the stomatal conductance of the wheat leaves by gently attaching the porometer sensor's head to the leaf. It was recorded in mmol m²s⁻¹. While, for the determination of the root biomass, root samples were collected washed with tap water to remove soil particles. After this, these are allowed to air dry and then kept in dry oven (101−1AB) for 65°C for 48−72 hours until all water content is removed, and only dry biomass is left behind. Then these were weighed on a digital weighing balance (NL, 7017 X) and recorded in kg ha⁻¹.

## Statistical analysis

The effects of copper and silicon nanoparticles on the inhibition zone of *Xtu*, BLS disease incidence, alterations in biochemical profile, and agronomic attributes of wheat plants were tested using analysis of variance (ANOVA) with LSD test at $p \leq 0.05$ employing Statistix 8.1 [34]. The graphical representation of the data was performed on Origin 2024b software.

## Results

### Characterization of nanoparticles

The sub-microscopic images of copper and silicon nanoparticles by scanning electron microscopy (SEM) showed that CuNPs exhibited semi-spherical morphology and SiNPs expressed spherical shape. While X-ray diffraction (XRD) analysis of CuNPs and SiNPs showed that NPs expressed crystalline nature exhibiting average particle size less than 100 nm as shown in Fig 1.

### Inhibition zone and disease incidence of bacterial leaf streak of wheat

Maximum inhibition zone was expressed by combination of CuNPs + SiNPs (19.06 mm), followed by CuNPs (14.14 mm) and SiNPs (10.56 mm) as compared to control. In case of interaction between T × C and T × D, CuNPs + SiNPs was the most effective followed by CuNPs and SiNPs in comparison with control (Fig 2 and 3A). Under greenhouse conditions, minimum disease incidence was expressed by CuNPs + SiNPs (14.75%), followed by CuNPs (34.92%) and SiNPs (41.11%) as compared to control. Interactions between T × C and T × D expressed that CuNPs + SiNPs exhibited minimum disease followed by CuNPs and SiNPs as compared to control (Fig 3B). While under field scale evaluation of nanoparticles, results revealed that CuNPs + SiNPs expressed minimum disease incidence followed by CuNPs and SiNPs as compared to control. T × D interaction with most effective concentration revealed that CuNPs + SiNPs expressed minimum disease incidence, followed by CuNPs and SiNPs (Fig 4).

### Estimation of alterations in the biochemical profiling of wheat

All enzymatic and non-enzymatic antioxidants activity was enhanced substantially after the application of copper and silicon nanoparticles alone and in hybrid. Application of most effective concentration of CuNPs + SiNPs enhanced the activity of SOD (3.09), POD (3.01), CAT (1.87), $H_2O_2$ (7.35), TPC (3.61), TSS (1.08) µg/g followed by CuNPs (2.56, 2.30, 1.58, 6.53, 3.03, 0.94) µg/g and SiNPs (2.09, 2.1.18, 1.28, 5.60, 2.10, 0.781) µg/g, respectively as compared to control (Fig 5).

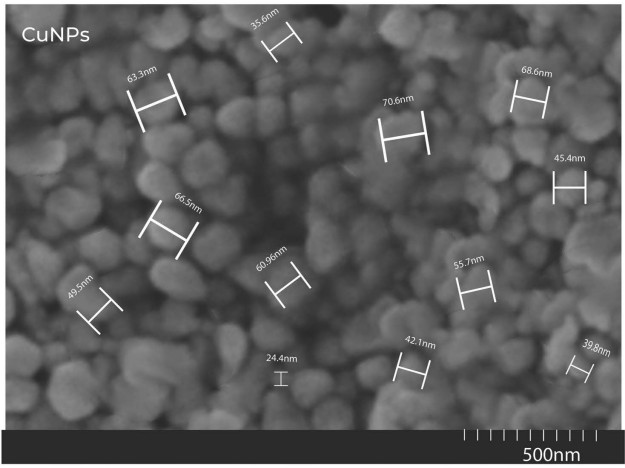 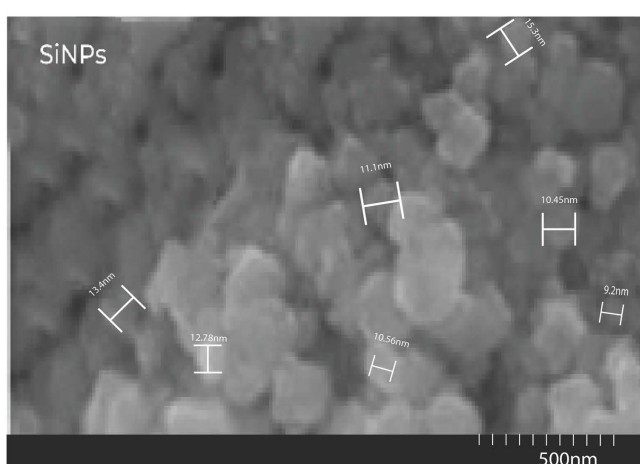

**Fig 1. SEM of green-synthesized copper and silicon nanoparticles.**

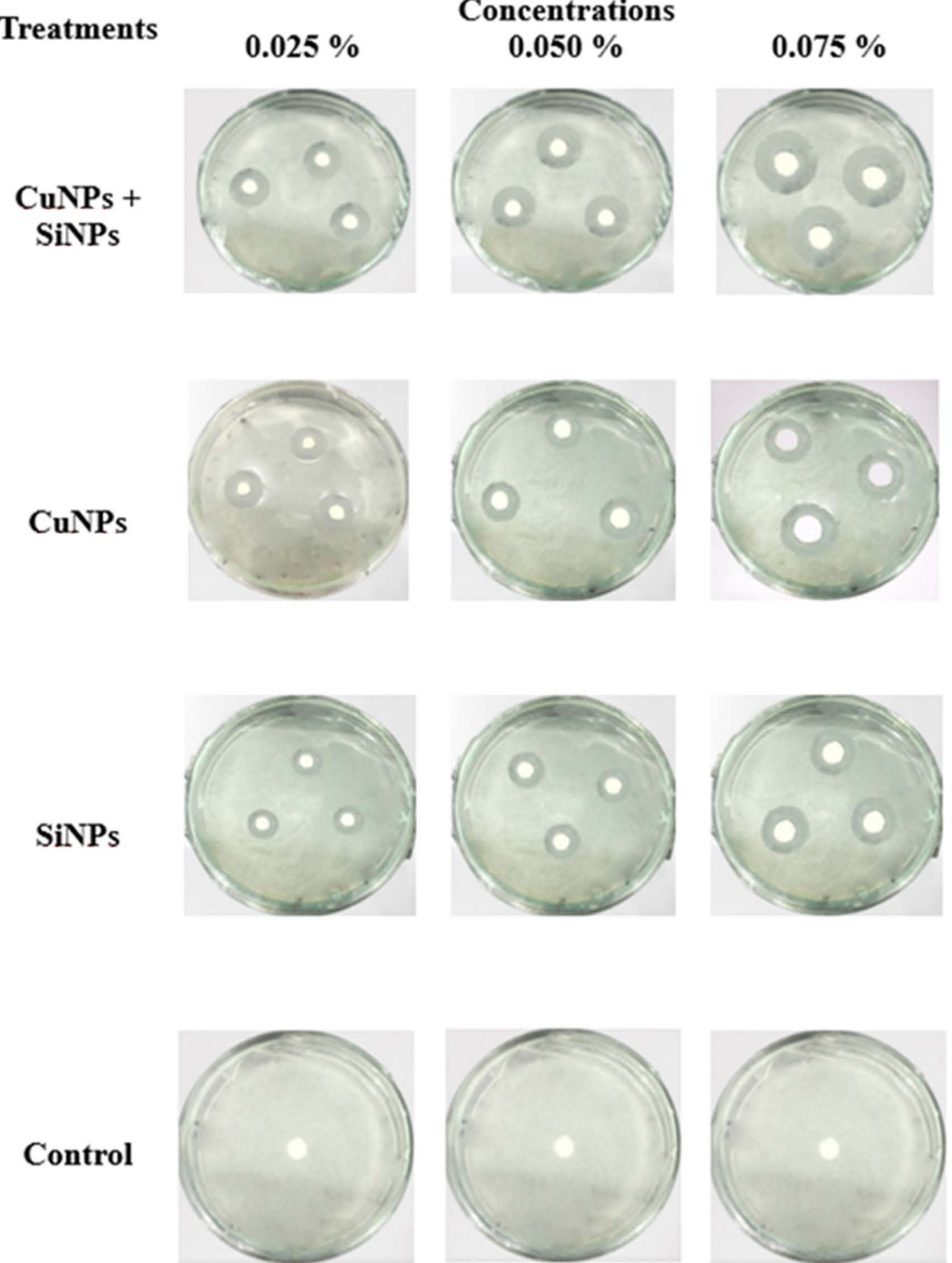

**Fig 2. Impact of copper and silicon nanoparticles on the Inhibition zone (mm) of *Xtu* in relation to different concentrations.**

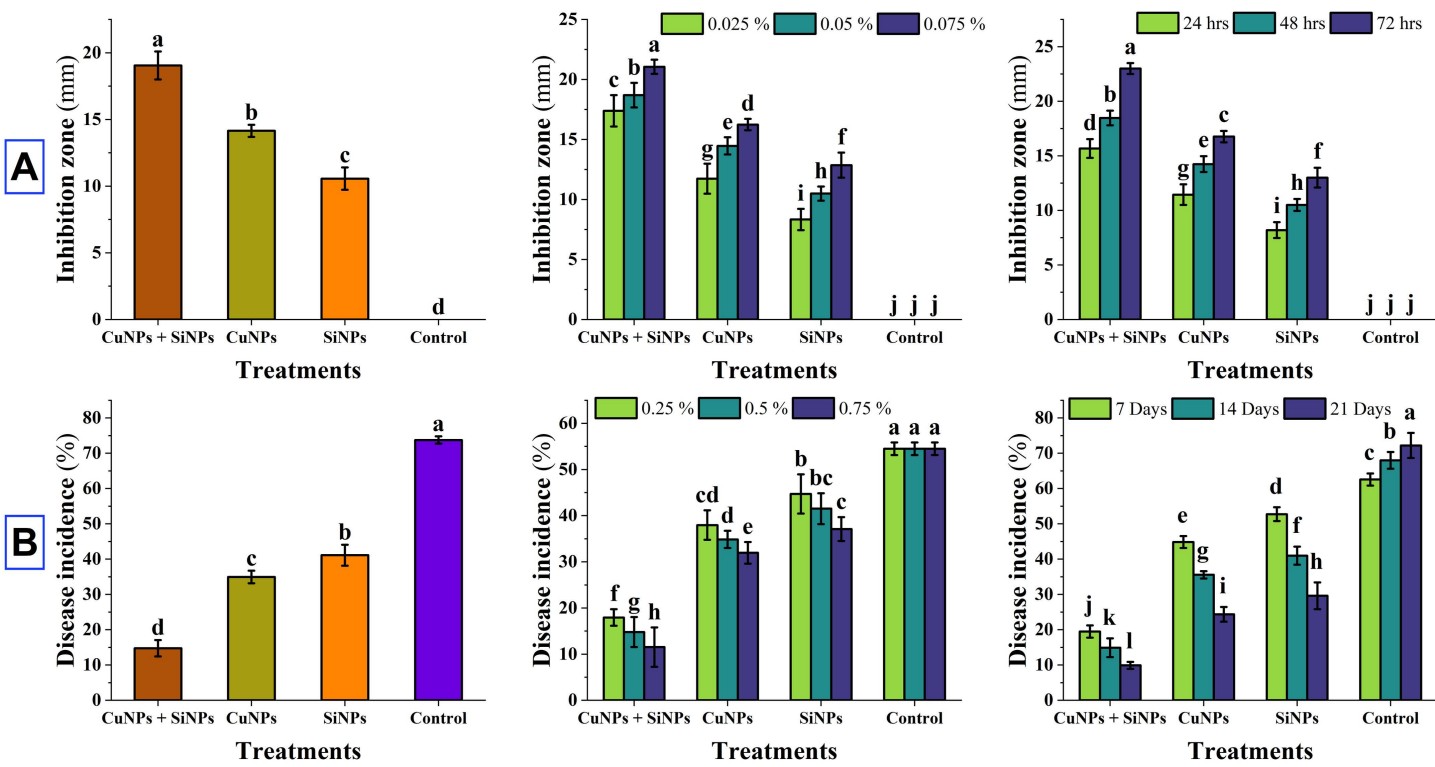

**Fig 3. Impact of copper and silicon nanoparticles on Inhibition zone (mm) (A), Disease incidence (%) under greenhouse (B).** Small letters represent the statistical significance using LSD test at $p \leq 0.05$.

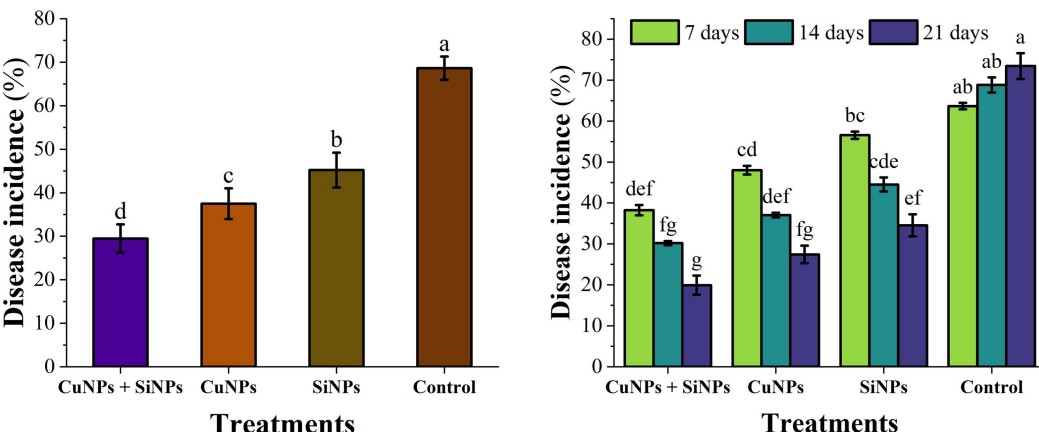

**Fig 4. Impact of copper and silicon nanoparticles on Disease incidence (%) under field conditions.** Small letters represent the statistical significance using LSD test at $p \leq 0.05$.

## Assessment of copper and silicon nanoparticles on the agronomic attributes of wheat

All the agronomic attributes of wheat were augmented substantially after the application of copper and silicon nanoparticles alone and in hybrid. Application of most effective concentration of CuNPs + SiNPs expressed the maximum root length (4.52 cm), shoot length (4.063 cm), chlorophyll contents (5.56 SPAD), spike length (15.98 cm),

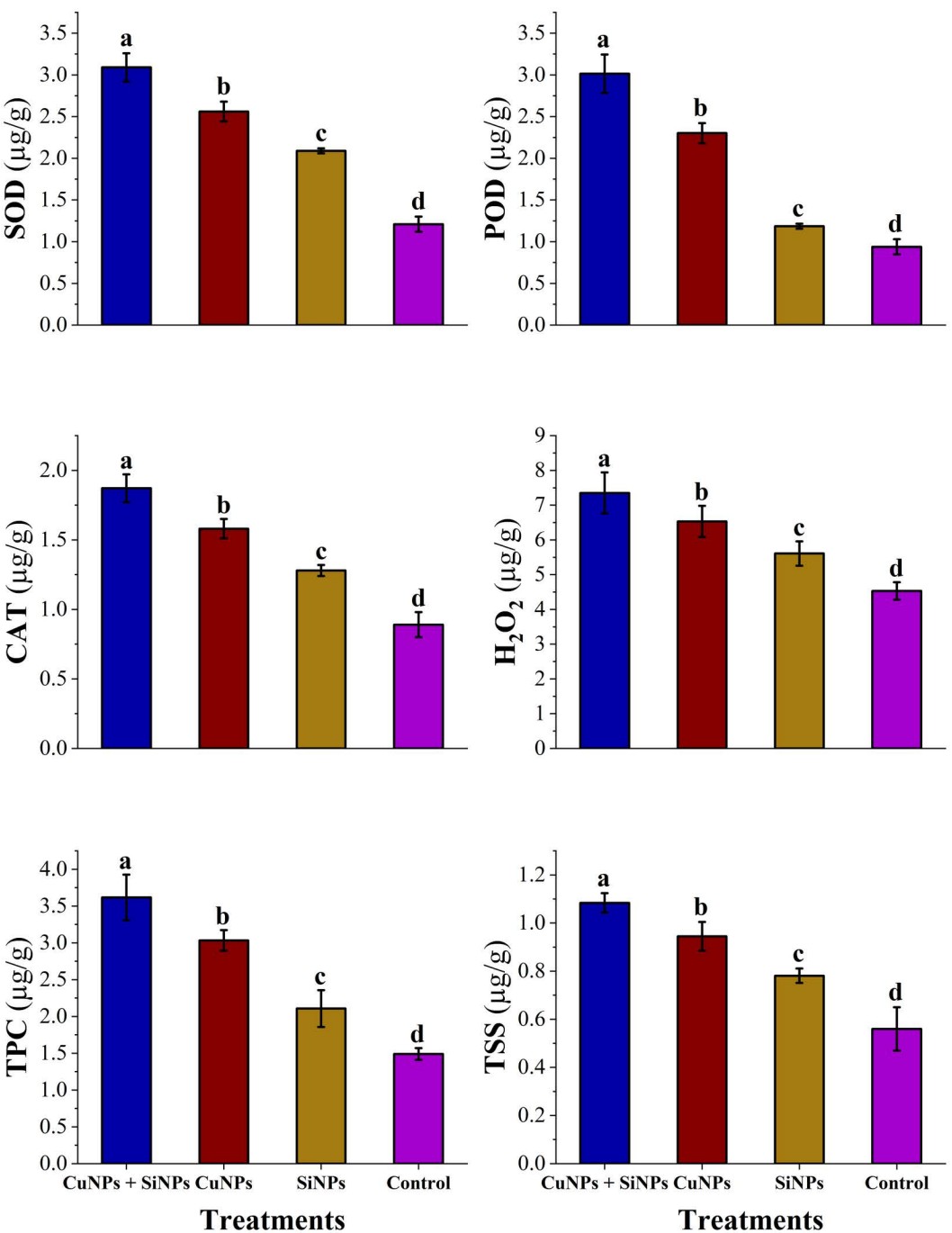

**Fig 5. Determination of SOD, POD, CAT, $H_2O_2$, TPC and TSS (µg/g) from treated and untreated wheat leaves after the application of nanoparticles.** Small letters represent the statistical significance using LSD test at $p \leq 0.05$.

spikelets per spike (19.88) and No. of grains per spike (31.97), followed by CuNPs and SiNPs in comparison with control as shown in Fig 6A. Similarly, CuNPs + SiNPs expressed the maximum 100 grains weight (17.07 g), plant height (81.46 cm), stomatal conductance (288.69 m.mol m$^{-2}$s$^{-1}$), and root biomass (492.15 kg/ha), followed by CuNPs and

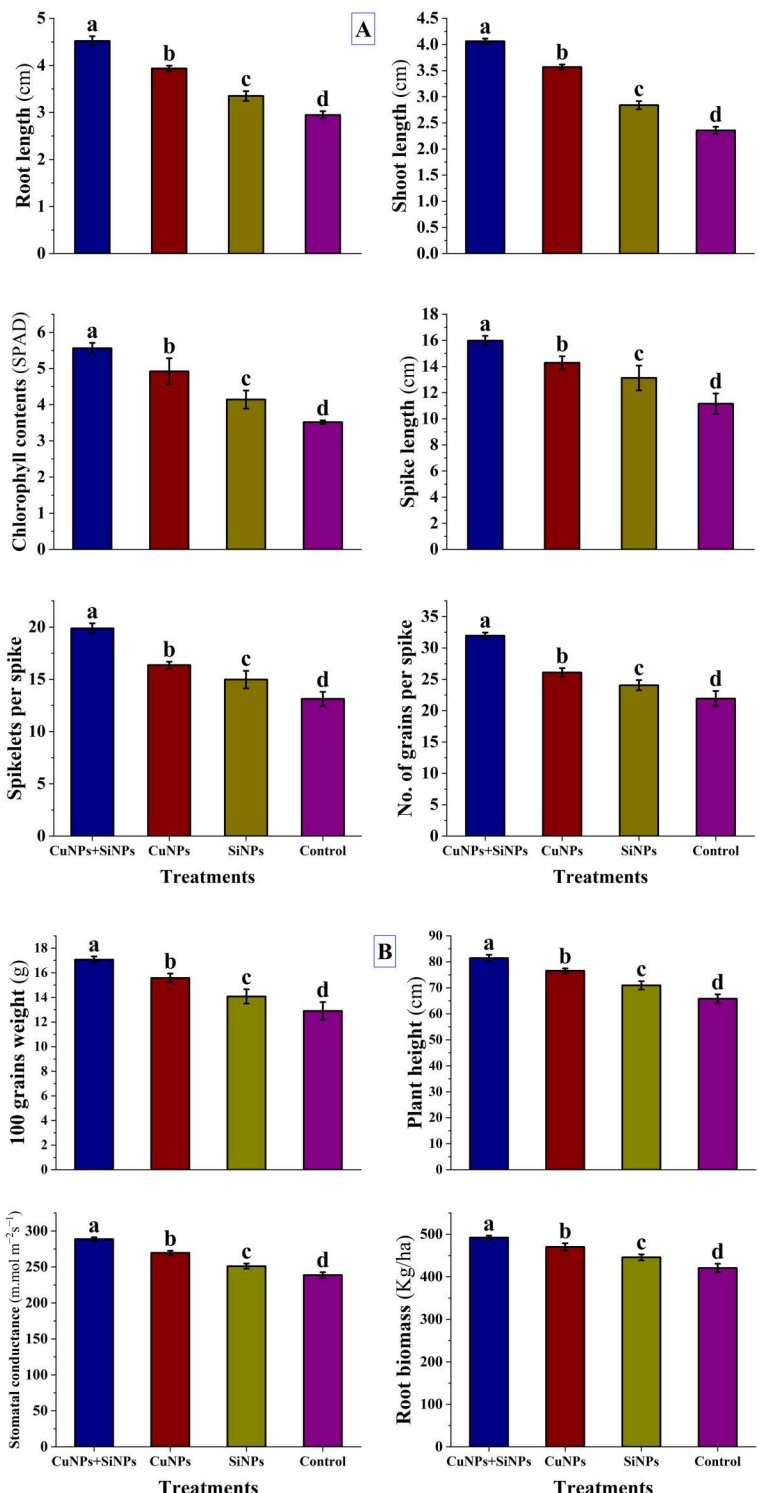

**Fig 6. A. Impact of copper and silicon nanoparticles on root length (cm), shoot length (cm), chlorophyll contents (SPAD), spike length (cm), spikelets per spike and No. of grains per spike.** Small letters correspond to the statistical significance using LSD test at $p \leq 0.05$. B. Impact of copper and silicon nanoparticles on 100 grains weight (g), plant height (cm), stomatal conductance (m.mol m$^{-2}$s$^{-1}$) and root biomass (kg/ha). Small letters correspond to the statistical significance using LSD test at $p \leq 0.05$.

SiNPs in comparison with control as shown in Fig 6B. Impact of nanoparticles on the yield-related aspects of plant are shown in Fig 7.

## Discussion

Excessive use of chemicals for disease management can disrupt plant functions and causes toxicity, leading to a range of health issues and organ impairments [14]. So, considering the hazardous effects of chemicals and the limitations of other control measures, advanced strategies such as green nanotechnology are essential for effective disease management [15]. The current research efforts were focused on the antibacterial efficacy of *Parthenium hysterophorus* mediated copper and silicon nanoparticles and their impacts on the biochemical profiling and yield-related attributes of the wheat for the effective management of BLS of wheat. The results revealed that hybrid (Cu + Si) nanoparticles expressed maximal inhibition zone under laboratory conditions, while minimum disease incidence in greenhouse and field conditions, followed by CuNPs and SiNPs as compared to control. Findings of the current research are encouraged by the outcomes of [35] who evaluated nanoparticles (Cu, MgO, ZnO) NPs, and concluded that CuNPs have the highest antibacterial potency against *Xanthomonas translucens* pv. *undulosa*. Results of this research are endorsed by the results of [26,36–38] who observed the antimicrobial activity of CuNPs. The expansion in the clear zone was observed in the petri plates by increasing time. This was due to the sustained release of active agents ($Cu^{2+}$), diffusion and interaction of NPs within the medium. The nanoparticles were characterized by employing UV-visible spectroscopy, scanning electron microscopy, and X-ray diffraction spectroscopy. The results revealed that copper nanoparticles exhibited the light brown to dark green color, semi-spherical shape with average particle size of 61.49 nm. These findings are in line with the work of [30,39,40], who performed the characterization of CuNPs synthesized from *Parthenium hysterophorus*. Similarly, silicon nanoparticles exhibited the spherical shape with the average particle size of 14.36 nm and these outcomes are in agreement with the findings of [41,42], who reported the average size and shape of the SiNPs.

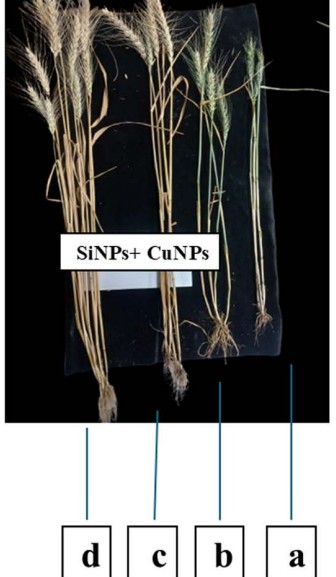
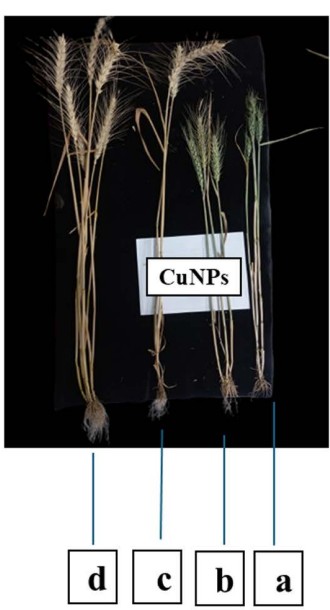
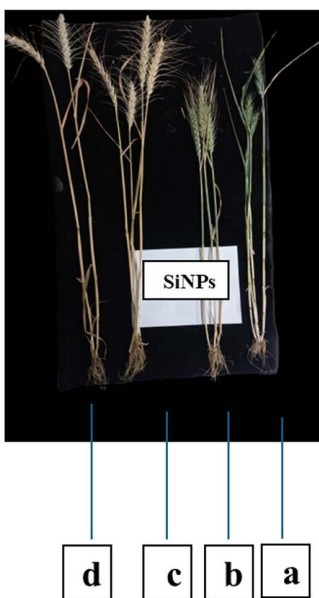

**Fig 7. Impact of nanoparticles on the yield-related attributes of wheat in relation to different concentrations 0.25 (b), 0.50 (c), 0.75% (d) of nanoparticles as compared to control (a).**

Copper and silicon nanoparticles have unique mechanisms for their uptake by the plants and suppressing the bacterial pathogens. In case of copper nanoparticles (CuNPs), they accumulate on the bacterial cell membrane and release $Cu^{2+}$ ions that interact with carboxyl as well as sulfhydryl groups in the peptidoglycan layer [43]. CuNPs enter the host plant through endocytosis [44]. Upon entry into the host plant, these $Cu^{2+}$ ions lead to the generation of reactive oxygen species (ROS), lipid peroxidation resulting in the interference with cell biomolecules, the protein functioning [45] and disrupt the ATP production, results in the disruption of the DNA, which leads to death of the bacterium [46].

Plants get SiNPs (5–20 mm) through endocytosis from roots and invade into the xylem tissues, translocated to different organs of the plant and stored in the shoots [47]. After accumulation, nanoparticles form a second layer of cuticle-Si, which strengthen the physical barriers against pathogenesis [48]. These changes in plants make it more difficult for the pathogens to invade and cause disease. Concurrently, SiNPs directly inhibit the pathogen growth by rupturing cell membranes which led to the spillage of cell contents resulting in cell death [25].

When plants are infected with *Xanthomonas* spp., then, a prominent alteration in the antioxidant defense enzymes like SOD, POD, CAT, and non-enzymatic antioxidants such as TPC, TSS, $H_2O_2$ was noted [49]. A surge in the induction of reactive oxygen species was experienced after the infection, which leads to an oxidative burst in the wheat plants [50]. Imbalance in the oxidants and antioxidants also resulted in oxidative stress which potentially damaged the plant cells. Higher levels of ROS resulted in the localized cell death and induce substantial tissue damage [51]. These combined consequences lead to the reduction in plant health and vigor which resulted in the limited growth and yield of plants. So, a most pressing need is to regulate the antioxidant defense system of wheat, so that they can exhibit the best growth and development [52]. In this study, the progressive alterations in the enzymatic antioxidants (SOD, POD, CAT) and non-enzymatic antioxidants (TPC, TSS, $H_2O_2$) from the treated and untreated wheat leaves after the application of copper and silicon nanoparticles alone and in hybrid was determined. The results revealed that Cu-Si hybrid nanoparticles expressed the maximum amounts of SOD, POD, CAT, TSS, TPC, $H_2O_2$ followed by copper nanoparticles and silicon nanoparticles in comparison with control. The application of the nanoparticles augmented the level of SOD, POD and CAT through controlled ROS production, directly interacting with enzymes, improving cellular environment, and upregulation of enzymatic genes expression which contributed to a more vigorous and effective antioxidant response, aiding the plants to survive under oxidative stress and make better its resilience to pathogens [42]. Superoxide dismutase (SOD), the first defense line against ROS like superoxide radicals ($O_2^-\bullet$) was activated in response to application of nanoparticles that converts the $O_2^-\bullet$ into hydrogen peroxide and oxygen ($2O_2^-\bullet + 2H^+ \rightarrow H_2O_2 + O_2$) [53,54]. Hydrogen peroxide ($H_2O_2$) is an important non-enzymatic antioxidant which acts as a signaling molecule as well as ROS. It is lethal to the pathogen at the optimum level but, when it exceeds, induces oxidative stress to the plants. Considering this oxidative stress, catalase activity (CAT) was triggered which converts the hydrogen peroxide from the dismutation ROS into water and oxygen ($2H_2O_2 \rightarrow 2H_2O + O_2$). It is capable of detoxifying millions of $H_2O_2$ molecules in seconds [55]. After detoxification of ROS, peroxidases (POD) reinforced the cell walls of the wheat plants acting as a physical barrier against pathogen's penetration into host plant. These results are in line with the study of [56], who reported the increase of antioxidant defense system of wheat plants after the application of nanoparticles. Phenolic compounds such as TPC are rich in hydroxyl groups that provide hydrogen atoms to counteract ROS like superoxide radicals and hydrogen peroxide efficiently and diminish oxidative stress in plants through reinforcing the cell walls, protecting lipid membranes, and induction of SAR [57–59]. Total soluble sugars (TSS) like glucose, sucrose, and fructose are essential sources of energy that are produced by metabolic processes such as citric acid cycle and glycolysis and are essential for the upregulation of enzymatic antioxidant defense system. These enzymes need ATP under stress where its demand is heightened for its effective activity [60]. The outcomes of the existent study aligned with the work of [61,62], who demonstrated that nanoparticles upregulated the activity of both enzymatic and non-enzymatic antioxidants of the plants. Additionally, [63–67] concluded that SiNPs exhibited rise in antioxidant defense system and noted significant enhancement in the growth of wheat plant by stimulating the antioxidant defense system through the application of silicon nanoparticles.

Nanoparticles allow precise release of bioactive compounds that not only suppress the negative effects of the phytopathogens but also promote the plant growth by inducing resistance in infected plants [29]. Current research effort was also directed to investigate the impact of *Parthenium hysterophorus* mediated Cu-Si hybrid nanotools towards the growth traits of wheat plant. The results of the study revealed that hybrid (CuNPs + SiNPs) nanoparticles expressed the maximum root length (cm), shoot length (cm), plant height (cm), spike length (cm), No. of grains per spike, spikelets per spike, 100 grains weight (g), chlorophyll contents (SPAD), root biomass (kg/ha) and stomatal conductance (m.mol $m^{-2}s^{-1}$), followed by CuNPs and SiNPs in comparison with control. Nanoparticles improved the chlorophyll contents of wheat by improving the uptake of the essential nutrients and then translocated to the leaves where synthesis of chlorophyll happened. Outcomes of this study are supported by the results of [68], who demonstrated that nanoparticles substantially enhanced the root length, shoot length and chlorophyll contents. Stomatal conductance was enhanced through the application of nanoparticles. Nanoparticles increased the water holding capacity of the wheat by strengthening the cell wall and increased root growth resulting in an increase in turgor pressure within guard cells (control stomatal opening) resulting in greater stomatal conductance. These findings are corroborated by the outcomes of [69], who revealed that stomatal conductance is elevated in the wheat plants treated with nanoparticles. The nanoparticles at lower concentrations significantly enhanced the overall plant height by improving the source-sink relationship, efficient nutrient and water uptake resulting in elevated photosynthetic activity. Similar results were observed by the findings of [70,71], who demonstrated that nanoparticles posed positive impacts on the plant height of wheat and rice. A surge in the shoot length, root length, no. of grains per spike was observed after the application of nanoparticles to wheat plants. This was due to the efficient nutrient uptake and stimulation of plant growth hormones such as gibberellins, cytokinin, and auxins responsible for the plant growth and these outcomes are endorsed by the results of [72,73]. At an optimum level, nanoparticles improved the agronomic parameters of wheat like number of grains per spike, root biomass, grains weight, spike length. These results are in line with the work of [74], who expressed that the application of nanoparticles enhanced the agronomic parameters of wheat. Moreover, existent results are aligned with outcomes of [56,67,75], who reported that nanoparticles significantly enhanced the agronomic attributes of wheat.

## Conclusion

Under laboratory conditions, the combination of CuNPs and SiNPs showed maximal inhibition zone against *Xanthomonas translucens* pv. *undulosa*. While, under field and greenhouse conditions, hybrid nanoparticles formulation demonstrated the lowest disease incidence compared to the individual applications of copper and silicon nanoparticles. Similarly, enzymatic and non-enzymatic defense system and yield-related attributes of wheat plant. Therefore, it is concluded that the hybrid form of copper and silicon nanoparticles is highly recommended to manage bacterial leaf streak of wheat and enhance antioxidant defense system and yield-related attributes of wheat plant effectively. Future research should focus on optimizing the hybrid nanoparticles, exploring the action mechanisms, assess impacts, and expand their application to crops and agroclimatic zones.

## Supporting information

**S1 Data.**
(XLSX)

**S2 Data.**
(XLSX)

**S3 Data.**
(XLSX)

## Acknowledgments

We authors extend our gratitude to the Molecular Bacteriology and Plant Pathogen Interaction Laboratory, Department of Plant Pathology UAF, Pakistan for providing the laboratory facilities necessary for conducting research.

## Author contributions

**Conceptualization:** Muhammad Atiq.

**Data curation:** Muhammad Jahanzaib Matloob, Hassan Zia.

**Formal analysis:** Rizwana Maqbool, Abdul Jabbar, Muhammad Usama.

**Investigation:** Waqas Ahmad.

**Methodology:** Jamil Shafi.

**Project administration:** Muhammad Atiq.

**Software:** Abdul Jabbar.

**Visualization:** Muhammad Jahanzaib Matloob.

**Writing – original draft:** Waqas Ahmad.

**Writing – review & editing:** Nasir Ahmed Rajput, Sohail Asad.

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
