## [Decision Letter · Decision Letter 0]

Dear Dr. Atiq,

We look forward to receiving your revised manuscript.

Kind regards,

Trung Quang Nguyen

Academic Editor

PLOS ONE

Journal Requirements:

Reviewers' comments:

Reviewer's Responses to Questions

**Comments to the Author**

1. Is the manuscript technically sound, and do the data support the conclusions?

Reviewer #1: Yes

Reviewer #2: Yes

Reviewer #3: Partly

Reviewer #4: Yes

2. Has the statistical analysis been performed appropriately and rigorously?

Reviewer #1: Yes

Reviewer #2: Yes

Reviewer #3: No

Reviewer #4: Yes

3. Have the authors made all data underlying the findings in their manuscript fully available?

Reviewer #1: Yes

Reviewer #2: Yes

Reviewer #3: No

Reviewer #4: Yes

4. Is the manuscript presented in an intelligible fashion and written in standard English?

Reviewer #1: No

Reviewer #2: Yes

Reviewer #3: Yes

Reviewer #4: No

Reviewer #1: Firstly, I would like to thank all those who contributed to this research paper,

1. The title needs to be clarified.

2. I highly recommend reviewing the grammar of the paper.

3. Introduction section;

- Line 50, start the Intro with reference No. [16], How??

- The references need to be rearranged.

4. Material and Methods section;

- Line 106 “ml” change to mL and so on.

- What specific synthesis techniques are most effective for combining silicon and copper nanoparticles?

- This method of combination isn’t mentioned.

- Why did you choose these three concentrations (0.025, 0.05, and 0.075 g).

- Line 182 “ntreated” changed to untreated.

5. Results

- Please, add the UV-vis and SEM figures.

Reviewer #2: The manuscript is well designed and well-prepared. Some comments are here and in the attached reviewed manuscript need to be addressed:

- throughout the manuscript check for English typos

- The title is too long, maximum should be 15 words

- the introduction need more recent literatures, I suggested some.

- the method should be referenced

- the figures and tables presenting the data very well, but the discussion should explain why the clear zone is increasing by the time

Statistical analysis should be done with two ways ANOVA to compare within the time interval

Reviewer #3: The authors will need to present the results for the characterization of the nanoparticles before application. The scanning electron microscope for the nanoparticles were not presented.

Experiments are better presented as three biological replicates rather than three replicates.

The zone of inhibitions represented on plate was not interpreted statistically.

The statistical application presented in the treatment figure 4 and figure 5 should be revisited.

Reviewer #4: Dear Authors,

Clarity and Structure: The manuscript presents an interesting study on the antibacterial efficacy of parthenium-mediated copper-silicon nanoparticles against bacterial leaf streak in wheat. However, the clarity of some sections could be improved. There are several formatting and spelling issues that need to be addressed throughout the manuscript. It is recommended to enhance the flow of the manuscript by ensuring that each section logically transitions to the next.

1. Abstract: The abstract is informative but could benefit from a more concise summary of the results and implications. Abbreviations (CuNPs, SiNPs, Cu-Si, etc.) should be expanded upon first use. Consider highlighting key findings and their significance in a more straightforward manner.

2. Introduction: While the introduction provides a solid background, it would be beneficial to include more recent references to strengthen the context of your research. The introduction is not up to date, and proper citations are missing. For example:

o Line 48: Pakistan ranks 8th in what? If you are referring to wheat, it is ranked 7th. Confirm this with the USDA and add the latest reference.

o Line 50: It appears that your references are not arranged correctly. Citations should start from 1, not 16. Follow the proper rules of citation.

Additionally, clearly stating the research objectives at the end of this section would provide a clearer focus for readers.

3. Methodology: The methods used for synthesizing and testing nanoparticles are well described. However, additional details regarding the controls used in experiments would enhance reproducibility. Clarify how concentrations were determined and provide justification for the selected dosages.

4. Results and Discussion: The results are presented clearly, but the discussion could be expanded to include comparisons with existing literature. Discussing potential mechanisms behind observed effects would provide deeper insights into your findings.

5. Figures: Ensure that all figures and tables are clearly labeled and referenced in the text. Some figures lack sufficient legends (e.g., Fig. 4, 5A, and 4B) explaining their significance. The photo quality is low; please upload high-resolution images (at least 300 dpi).

6. Tables: The table format is not suitable for publication. Please use the standard table format.

7. Conclusion: The conclusion summarizes the findings effectively but should also address future research directions or potential applications of your findings in agricultural practices.

8. Statistical Analysis: Please provide more details regarding the statistical methods used for data analysis, including the software and specific tests applied.

9. Side Effects: As parthenium is a causative agent of Parthenium dermatitis, what are the side effects of parthenium on wheat and ultimately on humans?

**Do you want your identity to be public for this peer review?** For information about this choice, including consent withdrawal, please see our Privacy Policy

Reviewer #1: **Yes: ** Shaimaa M. I. Alexeree

Reviewer #2: No

Reviewer #3: No

Reviewer #4: **Yes: ** Dr. Niaz Ali

---

## [Author Response · Author response to Decision Letter 1]

2 Dec 2024

Reviewer 1

Q1. The title needs to be clarified.

A.1 Title is modified.

Q2. I highly recommend reviewing the grammar of the paper

A.2 Grammatical Errors are corrected.

Q3. Introduction section;

(a) Line 50, start the Intro with reference No. [16], How?

(b) The references need to be rearranged

A.3 All the references in the text are rearranged according to your prescription

Q4. 4. Material and Methods section;

(a) Line 106 “ml” change to mL and so on.

(b) What specific synthesis techniques are most effective for combining silicon and copper nanoparticles?

(c) This method of combination isn’t mentioned.

(d) Why did you choose these three concentrations (0.025, 0.05, and 0.075 g).

(e) Line 182 “ntreated” changed to untreated.

A.4

(a) Changed to mL and so on.

(b) Simply the fine powder of both nanoparticles (Cu and Si) was mixed in a beaker and further used for their

assessment under laboratory, greenhouse and field conditions.

(c) It is stated in line 117.

(d) Nanoparticles cause toxicity to plants at higher concentrations. That’s why lower concentrations were most effective

for the best results.

(e) Incorporated.

Q5. Results;

(a) Please, add the UV-vis and SEM figures.

A.5. The SEM figures are attached. UV-vis pictures are not available.

Reviewer 2

Q1. Throughout the manuscript check for English typos

A1. Thoroughly proofread was done to remove any English typos errors

Q2. The title is too long, maximum should be 15 words

A2. Short title will not represent the focus of the research paper clearly,

that’s why the title is quite longer

Q3. the introduction need more recent literatures, I suggested some

A3. As per your suggestions, literature provided is recent. Kindly have a look

on it. Most of the references are above the year 2020 despite four

references

Q4. the method should be referenced

A4. Complete methodology is referenced

Q5. the figures and tables presenting the data very well, but the discussion

should explain why the clear zone is increasing by the time

A5. The suggestion is incorporated and explained why the clear zone is

increased with increase in time.

Q6. Statistical analysis should be done with two ways ANOVA to compare

within the time interval

A6. The statistical analysis was performed using three ways ANOVA to

compare it with time and concentration

Reviewer 3

Q1. The authors will need to present the results for the characterization of

the nanoparticles before application. The scanning electron microscope

for the nanoparticles were not presented

A1. SEM for the characterization of nanoparticles is represented in the

figures.

Q2. Experiments are better presented as three biological replicates rather

than three replicates

A2. Suggestions are incorporated

Q3. The zone of inhibitions represented on plate was not interpreted

statistically

A3. Incorporated.

Q4. The statistical application presented in the treatment figure 4 and figure

5 should be revisited

A5. Suggestions are corrected.

Reviewer 4

Q1. Abstract: The abstract is informative but could benefit from a more

concise summary of the results and implications. Abbreviations (CuNPs,

SiNPs, Cu-Si, etc.) should be expanded upon first use. Consider

highlighting key findings and their significance in a more straightforward

manner.

A1. Suggestions are incorporated

Q2. Introduction: While the introduction provides a solid background, it

would be beneficial to include more recent references to strengthen the

context of your research. The introduction is not up to date, and proper

citations are missing. For example:

(a) Line 48: Pakistan ranks 8th in what? If you are referring to wheat, it is

ranked 7th. Confirm this with the USDA and add the latest reference.

(b) Line 50: It appears that your references are not arranged correctly.

Citations should start from 1, not 16. Follow the proper rules of citation.

(c) Additionally, clearly stating the research objectives at the end of this

section would provide a clearer focus for readers

A2. The introduction is up to date. Kindly have a look on all references. Just

2-3 references are below to year 2020. All the remaining are of year

2020, 2021, 2022, 2023, and 2024.

a. It is confirmed that Pakistan stands at number 8th with respect to

wheat production globally.

b. All the references are rearranged as per their appearance in the text.

c. Objectives are clearly stated at the end of introduction.

Q3. Methodology: The methods used for synthesizing and testing

nanoparticles are well described. However, additional details regarding

the controls used in experiments would enhance reproducibility. Clarify

how concentrations were determined and provide justification for the

selected dosages.

A3. The control is referred to as distilled water during the whole experiment.

To get three concentrations (0.025, 0.05, and 0.075 %) of NPs, 0.025,

0.05, and 0.075 g of (Cu and Si) powder alone and in combination (Cu

+ Si) were added in bottle (250 mL) containing 100 mL distilled water

separately. The nanoparticles cause toxicity to plants if applied at higher

concentrations that’s why small dosages are recommended

Q4. Results and Discussion: The results are presented clearly, but the

discussion could be expanded to include comparisons with existing

literature. Discussing potential mechanisms behind observed effects

would provide deeper insights into your findings

A4. In the discussion, the current findings are supported by latest studies

and compared with the existing literature. The potential mechanism

behind the action of both copper and silicon nanoparticles is described

very well.

Q5. Figures: Ensure that all figures and tables are clearly labeled and

referenced in the text. Some figures lack sufficient legends (e.g., Fig. 4,

5A, and 4B) explaining their significance. The photo quality is low;

please upload high-resolution images (at least 300 dpi).

A5. The Fig. 4, 5A and 5B are representing the data very well and their

resolution is also at least 300 dpi

Q6. Tables: The table format is not suitable for publication. Please use the

standard table format.

A6. Table is removed from the manuscript emphasizing the formatting of the

manuscript.

Q7. Conclusion: The conclusion summarizes the findings effectively but

should also address future research directions or potential applications of

your findings in agricultural practices.

A7. Future research directions are updated.

Q8. Statistical Analysis: Please provide more details regarding the statistical

methods used for data analysis, including the software and specific

tests applied.

A8. Statistical analysis is performed by employing ANOVA (Analysis of

Variance) with LSD test at p ≤ 0.05 in software named as “Statistix 8.1”.

The graphical representation was performed on “Origin 2024b”.

Q9. Side Effects: As parthenium is a causative agent of Parthenium

dermatitis, what are the side effects of parthenium on wheat and

ultimately on humans?

A9. Parthenium dermatitis is one of the main constraints faced by human

health as a prolonged exposure to Parthenium hysterophorus L. It can

also cause respiratory risks like asthma and bronchitis

---

## [Decision Letter · Decision Letter 1]

Dear Dr. Atiq,

Thank you for submitting your manuscript to PLOS ONE. After careful consideration, we feel that it has merit but does not fully meet PLOS ONE’s publication criteria as it currently stands. Therefore, we invite you to submit a revised version of the manuscript that addresses the points raised during the review process.

We look forward to receiving your revised manuscript.

Kind regards,

Trung Quang Nguyen

Academic Editor

PLOS ONE

Journal Requirements:

Reviewers' comments:

Reviewer's Responses to Questions

**Comments to the Author**

Reviewer #2: (No Response)

Reviewer #4: (No Response)

2. Is the manuscript technically sound, and do the data support the conclusions?

Reviewer #2: Partly

Reviewer #4: Yes

3. Has the statistical analysis been performed appropriately and rigorously?

Reviewer #2: Yes

Reviewer #4: Yes

4. Have the authors made all data underlying the findings in their manuscript fully available?

Reviewer #2: No

Reviewer #4: Yes

5. Is the manuscript presented in an intelligible fashion and written in standard English?

Reviewer #2: Yes

Reviewer #4: Yes

Reviewer #2: The author did response to the comments but all the comments in the attached reviewed manuscript are missing.

Check and address all the comments

Reviewer #4: Comments to the Authors:

The authors have incorporated the suggestions. However, in some comments, the authors did not follow the instructions. For example:

1. Pakistan ranks 7th in wheat production instead of 8th (https://fas.usda.gov/data/production/commodity/0410000).

2. The wheat contribution to agricultural value added and GDP is incorrect. The authors must update their manuscript according to the recent reference (Investigating the Impact of Agricultural Credit on Wheat Farming: Evidence from Pakistan, https://doi.org/10.3390/agriculture14122200).

I advise the authors to review the above information and verify it with the latest online or published data (if available). The purpose of the revision is to improve and provide globally accepted and authenticated data for publication.

**Do you want your identity to be public for this peer review?** For information about this choice, including consent withdrawal, please see our Privacy Policy

Reviewer #2: No

Reviewer #4: **Yes: ** Dr. Niaz Ali

---

## [Author Response · Author response to Decision Letter 2]

9 Jan 2025

PLOS ONE Reviewer comments

Reviewer 1

Reviewer comments Responses

1. The title needs to be clarified. Incorporated

2. I highly recommend reviewing the grammar of the paper. Grammatical Errors are corrected.

3. Introduction section;

(a) Line 50, start the Intro with reference No. [16], How?

(b) The references need to be rearranged. All the references in the text are rearranged according to your prescription.

4. Material and Methods section;

(a) Line 106 “ml” change to mL and so on.

(b) What specific synthesis techniques are most effective for combining silicon and copper nanoparticles?

(c) This method of combination isn’t mentioned.

(d) Why did you choose these three concentrations (0.025, 0.05, and 0.075 g).

(e) Line 182 “ntreated” changed to untreated. a. Changed to mL and so on.

b. Simply the fine powder of both nanoparticles (Cu and Si) was mixed in a beaker and further used for their assessment under laboratory, greenhouse and field conditions.

c. It is stated in line 117.

d. Nanoparticles cause toxicity to plants at higher concentrations. That’s why lower concentrations were most effective for the best results.

e. Incorporated.

5. Results;

(a) Please, add the UV-vis and SEM figures. The SEM figures are attached. UV-vis pictures are not available.

Reviewer 2

Reviewer comments Responses

1. Throughout the manuscript check for English typos Thoroughly proofread was done to remove any English typos errors.

2. The title is too long, maximum should be 15 words. Short title will not represent the focus of the research paper clearly, that’s why the title is quite longer.

3. the introduction need more recent literatures, I suggested some. As per your suggestions, literature provided is recent. Kindly have a look on it. Most of the references are above the year 2020 despite four references.

4. the method should be referenced Complete methodology is referenced.

5. the figures and tables presenting the data very well, but the discussion should explain why the clear zone is increasing by the time The suggestion is incorporated and explained why the clear zone is increased with increase in time.

6. Statistical analysis should be done with two ways ANOVA to compare within the time interval The statistical analysis was performed using three ways ANOVA to compare it with time and concentration.

7. The author did response to the comments but all the comments in the attached reviewed manuscript are missing.

Check and address all the comments In this revision, all the relevant comments that were missed in the previous revision are addressed properly with comments section using track changes. Kindly check it.

Reviewer 3

Reviewer comments Responses

1. The authors will need to present the results for the characterization of the nanoparticles before application. The scanning electron microscope for the nanoparticles were not presented. SEM for the characterization of nanoparticles is represented in the figures.

2. Experiments are better presented as three biological replicates rather than three replicates. Suggestions are incorporated.

3. The zone of inhibitions represented on plate was not interpreted statistically. Incorporated.

4. The statistical application presented in the treatment figure 4 and figure 5 should be revisited. Suggestions are corrected.

Reviewer 4

Reviewer comments Responses

1. Abstract: The abstract is informative but could benefit from a more concise summary of the results and implications. Abbreviations (CuNPs, SiNPs, Cu-Si, etc.) should be expanded upon first use. Consider highlighting key findings and their significance in a more straightforward manner. Suggestions are incorporated.

2. Introduction: While the introduction provides a solid background, it would be beneficial to include more recent references to strengthen the context of your research. The introduction is not up to date, and proper citations are missing. For example:

(a) Line 48: Pakistan ranks 8th in what? If you are referring to wheat, it is ranked 7th. Confirm this with the USDA and add the latest reference.

(b) Line 50: It appears that your references are not arranged correctly. Citations should start from 1, not 16. Follow the proper rules of citation.

(c) Additionally, clearly stating the research objectives at the end of this section would provide a clearer focus for readers. The introduction is up to date. Kindly have a look on all references. Just 2-3 references are below to year 2020. All the remaining are of year 2020, 2021, 2022, 2023, and 2024.

a. It is confirmed that Pakistan stands at number 8th with respect to wheat production globally.

b. All the references are rearranged as per their appearance in the text.

c. Objectives are clearly stated at the end of introduction.

3. Methodology: The methods used for synthesizing and testing nanoparticles are well described. However, additional details regarding the controls used in experiments would enhance reproducibility. Clarify how concentrations were determined and provide justification for the selected dosages. The control is referred to as distilled water during the whole experiment. To get three concentrations (0.025, 0.05, and 0.075 %) of NPs, 0.025, 0.05, and 0.075 g of (Cu and Si) powder alone and in combination (Cu + Si) were added in bottle (250 mL) containing 100 mL distilled water separately. The nanoparticles cause toxicity to plants if applied at higher concentrations that’s why small dosages are recommended.

4. Results and Discussion: The results are presented clearly, but the discussion could be expanded to include comparisons with existing literature. Discussing potential mechanisms behind observed effects would provide deeper insights into your findings. In the discussion, the current findings are supported by latest studies and compared with the existing literature. The potential mechanism behind the action of both copper and silicon nanoparticles is described very well.

5. Figures: Ensure that all figures and tables are clearly labeled and referenced in the text. Some figures lack sufficient legends (e.g., Fig. 4, 5A, and 4B) explaining their significance. The photo quality is low; please upload high-resolution images (at least 300 dpi). The Fig. 4, 5A and 5B are representing the data very well and their resolution is also at least 300 dpi.

6. Tables: The table format is not suitable for publication. Please use the standard table format. Table is removed from the manuscript emphasizing the formatting of the manuscript.

7. Conclusion: The conclusion summarizes the findings effectively but should also address future research directions or potential applications of your findings in agricultural practices. Future research directions are updated.

8. Statistical Analysis: Please provide more details regarding the statistical methods used for data analysis, including the software and specific tests applied. Statistical analysis is performed by employing ANOVA (Analysis of Variance) with LSD test at p ≤ 0.05 in software named as “Statistix 8.1”. The graphical representation was performed on “Origin 2024b”.

9. Side Effects: As parthenium is a causative agent of Parthenium dermatitis, what are the side effects of parthenium on wheat and ultimately on humans? Parthenium dermatitis is one of the main constraints faced by human health as a prolonged exposure to Parthenium hysterophorus L. It can also cause respiratory risks like asthma and bronchitis.

10. The authors have incorporated the suggestions. However, in some comments, the authors did not follow the instructions. For example:

1. Pakistan ranks 7th in wheat production instead of 8th (https://fas.usda.gov/data/production/commodity/0410000).

2. The wheat contribution to agricultural value added and GDP is incorrect. The authors must update their manuscript according to the recent reference (Investigating the Impact of Agricultural Credit on Wheat Farming: Evidence from Pakistan, https://doi.org/10.3390/agriculture14122200).

I advise the authors to review the above information and verify it with the latest online or published data (if available). The purpose of the revision is to improve and provide globally accepted and authenticated data for publication. All the comments raised during previous review were addressed in a scientific way.

1. This comment was also suggested to review and confirm the status of the Pakistan ranking for wheat production. It is confirmed by the data of FAOSTAT-2022 where it is clearly written that Pakistan ranked eighth in wheat production with respect to the globe. You can confirm this with this link of FAOSTAT

(https://www.fao.org/faostat/en/#rankings/countries_by_commodity). Kindly check it. It will represent that Pakistan is ranked eighth in production of wheat.

2. The agricultural value added by the wheat and its role in GDP of Pakistan is confirmed with the official economic survey of Pakistan under agriculture section. Kindly confirm this using the given link to Economic survey of Pakistan (https://www.finance.gov.pk/survey/chapter_24/2_agriculture.pdf).

---

## [Decision Letter · Decision Letter 2]

Dear Dr. Atiq,

Thank you for submitting your manuscript to PLOS ONE. After careful consideration, we feel that it has merit but does not fully meet PLOS ONE’s publication criteria as it currently stands. Therefore, we invite you to submit a revised version of the manuscript that addresses the points raised during the review process.

We look forward to receiving your revised manuscript.

Kind regards,

Trung Quang Nguyen

Academic Editor

PLOS ONE

Journal Requirements:

Reviewers' comments:

Reviewer's Responses to Questions

**Comments to the Author**

Reviewer #2: (No Response)

2. Is the manuscript technically sound, and do the data support the conclusions?

Reviewer #2: Yes

3. Has the statistical analysis been performed appropriately and rigorously?

Reviewer #2: Yes

4. Have the authors made all data underlying the findings in their manuscript fully available?

Reviewer #2: Yes

5. Is the manuscript presented in an intelligible fashion and written in standard English?

Reviewer #2: Yes

Reviewer #2: The author responded partially to the comments, as there are several comments in the attached reviewed manuscript that are not yet addressed in the submitted revised manuscript

**Do you want your identity to be public for this peer review?** For information about this choice, including consent withdrawal, please see our Privacy Policy

Reviewer #2: No

---

## [Author Response · Author response to Decision Letter 3]

28 Feb 2025

Reviewer1

1. The title needs to be clarified.

A. Incorporated

2. I highly recommend reviewing the grammar of the paper.

A. Grammatical Errors are corrected.

3. 1. Introduction section;

(a) Line 50, start the Intro with reference No. [16], How?

(b) The references need to be rearranged.

A. All the references in the text are rearranged according to your prescription.

4. 4. Material and Methods section;

(a) Line 106 “ml” change to mL and so on.

(b) What specific synthesis techniques are most effective for combining silicon and copper nanoparticles?

(c) This method of combination isn’t mentioned.

(d) Why did you choose these three concentrations (0.025, 0.05, and 0.075 g).

(e) Line 182 “ntreated” changed to untreated

A. a. Changed to mL and so on.

b. Simply the fine powder of both nanoparticles (Cu and Si) was mixed in a beaker and further used for their assessment under laboratory, greenhouse and field conditions.

c. It is stated in line 117.

d. Nanoparticles cause toxicity to plants at higher concentrations. That’s why lower concentrations were most effective for the best results.

e. Incorporated.

5. 5. Results;

Please, add the UV-vis and SEM figures

A. The SEM figures are attached. UV-vis pictures are not available.

Reviewer 2

1. Throughout the manuscript check for English typos

A. Thoroughly proofread was done to remove any English typos errors.

2.The title is too long, maximum should be 15 words

A. Short title will not represent the focus of the research paper clearly, that’s why the title is quite longer.

3. the introduction needs more recent literatures, I suggested some.

A. As per your suggestions, literature provided is recent. Kindly have a look on it. Most of the references are above the year 2020 despite four references.

4. the method should be referenced

A. Complete methodology is referenced.

5. the figures and tables presenting the data very well, but the discussion should explain why the clear zone is increasing by the time

A. The suggestion is incorporated and explained why the clear zone is increased with increase in time.

6. Statistical analysis should be done with two ways ANOVA to compare within the time interval

A. The statistical analysis was performed using three ways ANOVA to compare it with time and concentration.

7. The author did response to the comments but all the comments in the attached reviewed manuscript are missing.

Check and address all the comments

8. 1. The author responded partially to the comments, as there are several comments in the attached reviewed manuscript that are not yet addressed in the submitted revised manuscript

A. In this revision, all the relevant comments missing in previous revision are addressed properly with comments section.

Reviewer 3.

1. The authors will need to present the results for the characterization of the nanoparticles before application. The scanning electron microscope for the nanoparticles were not presented.

A. SEM for the characterization of nanoparticles is represented in the figures.

2.Experiments are better presented as three biological replicates rather than three replicates.

A. Suggestions are incorporated.

3. The zone of inhibitions represented on plate was not interpreted statistically.

A. Incorporated.

4. The statistical application presented in the treatment figure 4 and figure 5 should be revisited.

A. Suggestions are incorporated.

A. In this revision, all the relevant comments that were missed in the previous revision are addressed properly with comments section using track changes. Kindly check it.

Reviewer 4.

1. Abstract: The abstract is informative but could benefit from a more concise summary of the results and implications. Abbreviations (CuNPs, SiNPs, Cu-Si, etc.) should be expanded upon first use. Consider highlighting key findings and their significance in a more straightforward manner.

A. Suggestions are incorporated.

2. Introduction: While the introduction provides a solid background, it would be beneficial to include more recent references to strengthen the context of your research. The introduction is not up to date, and proper citations are missing. For example:

(a) Line 48: Pakistan ranks 8th in what? If you are referring to wheat, it is ranked 7th. Confirm this with the USDA and add the latest reference.

(b) Line 50: It appears that your references are not arranged correctly. Citations should start from 1, not 16. Follow the proper rules of citation.

(c) Additionally, clearly stating the research objectives at the end of this section would provide a clearer focus for readers.

A. The introduction is up to date. Kindly have a look on all references. Just 2-3 references are below to year 2020. All the remaining are of year 2020, 2021, 2022, 2023, and 2024.

a. It is confirmed that Pakistan stands at number 8th with respect to wheat production globally.

b. All the references are rearranged as per their appearance in the text.

c. Objectives are clearly stated at the end of introduction.

3. Methodology: The methods used for synthesizing and testing nanoparticles are well described. However, additional details regarding the controls used in experiments would enhance reproducibility. Clarify how concentrations were determined and provide justification for the selected dosages.

A. The control is referred to as distilled water during the whole experiment. To get three concentrations (0.025, 0.05, and 0.075 %) of NPs, 0.025, 0.05, and 0.075 g of (Cu and Si) powder alone and in combination (Cu + Si) were added in bottle (250 mL) containing 100 mL distilled water separately. The nanoparticles cause toxicity to plants if applied at higher concentrations that’s why small dosages are recommended.

4. Results and Discussion: The results are presented clearly, but the discussion could be expanded to include comparisons with existing literature. Discussing potential mechanisms behind observed effects would provide deeper insights into your findings.

A. In the discussion, the current findings are supported by latest studies and compared with the existing literature. The potential mechanism behind the action of both copper and silicon nanoparticles is described very well.

5. Figures: Ensure that all figures and tables are clearly labeled and referenced in the text. Some figures lack sufficient legends (e.g., Fig. 4, 5A, and 4B) explaining their significance. The photo quality is low; please upload high-resolution images (at least 300 dpi).

A. The Fig. 4, 5A and 5B are representing the data very well and their resolution is also at least 300 dpi.

6. Tables: The table format is not suitable for publication. Please use the standard table format.

A. Table is removed from the manuscript emphasizing the formatting of the manuscript.

7. Conclusion: The conclusion summarizes the findings effectively but should also address future research directions or potential applications of your findings in agricultural practices.

A. Future research directions are updated.

8. Statistical Analysis: Please provide more details regarding the statistical methods used for data analysis, including the software and specific tests applied.

A. Statistical analysis is performed by employing ANOVA (Analysis of Variance) with LSD test at p ≤ 0.05 in software named as “Statistix 8.1”. The graphical representation was performed on “Origin 2024b”.

9. Side Effects: As parthenium is a causative agent of Parthenium dermatitis, what are the side effects of parthenium on wheat and ultimately on humans?

A. Parthenium dermatitis is one of the main constraints faced by human health as a prolonged exposure to Parthenium hysterophorus L. It can also cause respiratory risks like asthma and bronchitis.

10. 10. The authors have incorporated the suggestions. However, in some comments, the authors did not follow the instructions. For example:

1. Pakistan ranks 7th in wheat production instead of 8th (https://fas.usda.gov/data/production/commodity/0410000).

2. The wheat contribution to agricultural value added and GDP is incorrect. The authors must update their manuscript according to the recent reference (Investigating the Impact of Agricultural Credit on Wheat Farming: Evidence from Pakistan, https://doi.org/10.3390/agriculture14122200).

I advise the authors to review the above information and verify it with the latest online or published data (if available). The purpose of the revision is to improve and provide globally accepted and authenticated data for publication

A. All the comments raised during previous review were addressed in a scientific way.

1. This comment was also suggested to review and confirm the status of the Pakistan ranking for wheat production. It is confirmed by the data of FAOSTAT-2022 where it is clearly written that Pakistan ranked eighth in wheat production with respect to the globe. You can confirm this with this link of FAOSTAT

(https://www.fao.org/faostat/en/#rankings/countries_by_commodity). Kindly check it. It will represent that Pakistan is ranked eighth in production of wheat.

2. The agricultural value added by the wheat and its role in GDP of Pakistan is confirmed with the official economic survey of Pakistan under agriculture section. Kindly confirm this using the given link to Economic survey of Pakistan (https://www.finance.gov.pk/survey/chapter_24/2_agriculture.pdf

---

## [Decision Letter · Decision Letter 3]

Dear Dr.  Atiq ,

Thank you for submitting your manuscript to PLOS ONE. After careful consideration, we feel that it has merit but does not fully meet PLOS ONE’s publication criteria as it currently stands. Therefore, we invite you to submit a revised version of the manuscript that addresses the points raised during the review process.

We look forward to receiving your revised manuscript.

Kind regards,

Trung Quang Nguyen

Academic Editor

PLOS ONE

Journal Requirements:

Reviewers' comments:

Reviewer's Responses to Questions

**Comments to the Author**

Reviewer #2: All comments have been addressed

2. Is the manuscript technically sound, and do the data support the conclusions?

Reviewer #2: Yes

3. Has the statistical analysis been performed appropriately and rigorously?

Reviewer #2: Yes

4. Have the authors made all data underlying the findings in their manuscript fully available?

Reviewer #2: Yes

5. Is the manuscript presented in an intelligible fashion and written in standard English?

Reviewer #2: Yes

Reviewer #2: The author responded to all previous comments, but still some more comments in the attached pdf need to be addressed.

**Do you want your identity to be public for this peer review?** For information about this choice, including consent withdrawal, please see our Privacy Policy

Reviewer #2: No

---

## [Author Response · Author response to Decision Letter 4]

16 Apr 2025

Reviewer comments Responses

Replace this reference : Shafqat U, Maqsood A, Ishfaq A, Mustafa S, Rasheed Y, Mahmood F, et al. Green nanotechnology for plant bacterial diseases management in cereal crops: a review on metal-based nanoparticles. Notulae Bot Horti Agrobotan Cluj-Napoca. 2023 Sep 27;51(3): 13333. doi: https://doi.org/10.15835/nbha51313333

The highlighted reference is Replaced with:

Francis DV, Abdalla AK, Mahakham W, Sarmah AK, Ahmed ZF. Interaction of plants and metal nanoparticles: Exploring its molecular mechanisms for sustainable agriculture and crop improvement. Environ Int. 2024 Aug; 108859. doi: https://doi.org/10.1016/j.envint.2024.108859

Replace this reference : Ameta SK, Ameta SC. Eco-friendly approaches of using weeds for sustainable plant growth and production. In: Husen A, editor. Plant performance under environmental stress. Springer, Singapore; 2021. pp. 559-592. doi: https://doi.org/10.1007/978-3-030-78521-5

The highlighted reference is Replaced with:

Francis DV, Abdalla AK, Mahakham W, Sarmah AK, Ahmed ZF. Interaction of plants and metal nanoparticles: Exploring its molecular mechanisms for sustainable agriculture and crop improvement. Environ Int. 2024 Aug; 108859. doi: https://doi.org/10.1016/j.envint.2024.108859

Replace this reference : Javaid A, Munir R, Khan IH, Shoaib A. Control of the chickpea blight, Ascochyta rabiei, with the weed plant, Withania somnifera. Egypt J Biol Pest Cont. 2020 Dec;30: 114. doi: https://doi.org/10.1186/s41938-020-00315-z

The highlighted reference is Replaced with

Asif A, Ali M, Qadir M, Karthikeyan R, Singh Z, Khangura R, Di Gioia F, Ahmed ZF. Enhancing crop resilience by harnessing the synergistic effects of biostimulants against abiotic stress. Front Plant Sci. 2023 Dec; 14: 1276117. doi: https://doi.org/10.3389/fpls.2023.1276117

---

## [Decision Letter · Decision Letter 4]

Dear Dr. Atiq,

Thank you for submitting your manuscript to PLOS ONE. After careful consideration, we feel that it has merit but does not fully meet PLOS ONE’s publication criteria as it currently stands. Therefore, we invite you to submit a revised version of the manuscript that addresses the points raised during the review process.

We look forward to receiving your revised manuscript.

Kind regards,

Trung Quang Nguyen

Academic Editor

PLOS ONE

Journal Requirements:

Reviewers' comments:

Reviewer's Responses to Questions

**Comments to the Author**

Reviewer #2: All comments have been addressed

2. Is the manuscript technically sound, and do the data support the conclusions?

Reviewer #2: Yes

3. Has the statistical analysis been performed appropriately and rigorously?

Reviewer #2: Yes

4. Have the authors made all data underlying the findings in their manuscript fully available?

Reviewer #2: Yes

5. Is the manuscript presented in an intelligible fashion and written in standard English?

Reviewer #2: Yes

Reviewer #2: The manuscript still need modifications. The comments in the attached reviewed manuscript need to be addressed to improve the manuscript.

**Do you want your identity to be public for this peer review?** For information about this choice, including consent withdrawal, please see our Privacy Policy

Reviewer #2: No

---

## [Author Response · Author response to Decision Letter 5]

22 May 2025

Reviewer comments Responses

1. The title is too long, max 15 words " RESILIENCE OF PARTHENIUM HYSTEROPHORUS-DERIVED COPPER-SILICON HYBRID NANOTOOLS AGAINST BACTERIAL LEAF STREAK AND THEIR IMPACT ON BIOCHEMICAL AND AGRONOMIC TRAITS OF WHEAT

A. Multimodal Impact Of Copper-Silicon Hybrid Nanotools towards Bacterial Leaf Streak , Wheat Biochemistry And Productivity Parameters

2. Replace Ref No. 5: Hussain D, Asrar M, Khalid B, Hafeez F, Saleem M, Akhter M, et al. Insect pests of economic importance attacking wheat crop (Triticum aestivum L.) in Punjab, Pakistan. Int J Trop Insect Sci. 2022 Feb;42(1): 9-20. doi: https://doi.org/10.1007/s42690-021-00574-9

A. Kaur N, Muslim Q, Dali VF, Anshu A, Siddharth T, Zienab FA. CRISPR/Cas9: A sustainable technology to enhance climate resilience in major staple crops. Front Genome Ed. 2025 March 18; 7: 1533197. https://doi.org/10.3389/fgeed.2025.1533197

3. Replace Ref No. 10: Ledman KE, Curland RD, Ishimaru CA, Dill-Macky R. Xanthomonas translucens pv. undulosa identified on common weedy grasses in naturally infected wheat fields in Minnesota. Phytopathol. 2021 Jul 30;111(7): 1114-1121. doi: https://doi.org/10.1094/PHYTO-08-20-0337-R

A. Hussain S, Aqleem A, Maratab A, Israt J, Muhammad J, Muhammad A.A, et al. Diversity of Alternaria-derived toxins and their toxicodynamic and toxicokinetic characteristics in the food chain. Food Front. 2025; 6(1): 185-217. https://doi.org/10.1002/fft2.507

4. Replace Ref No. 17: Shafqat U, Maqsood A, Ishfaq A, Mustafa S, Rasheed Y, Mahmood F, et al. Green nanotechnology for plant bacterial diseases management in cereal crops: a review on metal-based nanoparticles. Notulae Bot Horti Agrobotan Cluj-Napoca. 2023 Sep 27;51(3): 13333. doi: https://doi.org/10.15835/nbha51313333

A. Asif A, Maratab A, Muslim Q, Rajmohan K, Zora S, Ravjit K, et al. Enhancing crop resilience by harnessing the synergistic effects of biostimulants against abiotic stress. Front Plant Sci. 2023 Dec 18; 14: 1276117. https://doi.org/10.3389/fpls.2023.1276117

5. Replace Ref No. 22: Kashyap P, Shirkot P, Das R, Pandey H, Singh D. Biosynthesis and characterization of copper nanoparticles from Stenotrophomonas maltophilia and its effect on plant pathogens and pesticide degradation. J Agri Food Res. 2023 Sep 1;13: 100654. doi: https://doi.org/10.1016/j.jafr.2023.100654

A. Francis DV, Abdalla AK, Mahakham W, Sarmah AK, Ahmed ZF. Interaction of plants and metal nanoparticles: Exploring its molecular mechanisms for sustainable agriculture and crop improvement. Environ Int. 2024 Aug; 190: 108859. doi: https://doi.org/10.1016/j.envint.2024.108859

6. Replace Ref No. 27: Ali S, Ulhassan Z, Shahbaz H, Kaleem Z, Yousaf MA, Ali S, et al. Magnesium oxide nanoparticles as novel sustainable approach in enhancing crop tolerance to abiotic and biotic stresses. Environ Sci: Nano. 2024 Jul 5;11(8): 3250-3267. doi: https://doi.org/10.1039/D4EN00417E

A. Francis DV, Anam A, Zienab FRA. Nanoparticle-enhanced plant defense mechanisms harnessed by nanotechnology for sustainable crop protection. In: Masudulla K, Jen-Tsung C, editor. Nanoparticles in plant biotic stress management. Springer Nature; 2024 June 08. pp. 451-491. doi: https://doi.org/10.1007/978-981-97-0851-2_19

6. Replace Ref No. 30: Kralova K, Jampilek J. Applications of nanomaterials in plant disease management and protection. In: Ingle AP, editor. Nanotechnology in agriculture and agroecosystems. Micro Nano Tech. 2023 Jan 1: 239-296. doi: https://doi.org/10.1016/B978-0-323-99446-0.00013-1

A. Francis DV, Abdul S, Abdel-Hamid IM, Abdelmoneim KA, Zienab FRA. Optimizing germination conditions of Ghaf seed using ZnO nanoparticle priming through Taguchi method analysis. Sci Rep. 2024 July 10;14: 15946. doi: https://doi.org/10.1038/s41598-024-67025-6

---

## [Decision Letter · Decision Letter 5]

MULTIMODAL IMPACT OF COPPER-SILICON HYBRID NANOTOOLS TOWARDS BACTERIAL LEAF STREAK, WHEAT BIOCHEMISTRY AND PRODUCTIVITY PARAMETERS

PONE-D-24-41835R5

Dear Dr. Muhammad Atiq,

We’re pleased to inform you that your manuscript has been judged scientifically suitable for publication and will be formally accepted for publication once it meets all outstanding technical requirements.

Kind regards,

Trung Quang Nguyen

Academic Editor

PLOS ONE

Additional Editor Comments (optional):

Reviewers' comments:

Reviewer's Responses to Questions

**Comments to the Author**

Reviewer #2: All comments have been addressed

2. Is the manuscript technically sound, and do the data support the conclusions?

Reviewer #2: Yes

3. Has the statistical analysis been performed appropriately and rigorously?

Reviewer #2: Yes

4. Have the authors made all data underlying the findings in their manuscript fully available?

Reviewer #2: Yes

5. Is the manuscript presented in an intelligible fashion and written in standard English?

Reviewer #2: (No Response)

Reviewer #2: The author responded to all comments. The manuscript can be accepted if there is no negative report from other reviewers.

**Do you want your identity to be public for this peer review?** For information about this choice, including consent withdrawal, please see our Privacy Policy

Reviewer #2: No

---

## [Editor Report · Acceptance letter]

PONE-D-24-41835R5

PLOS ONE

Dear Dr. Atiq,

I'm pleased to inform you that your manuscript has been deemed suitable for publication in PLOS ONE. Congratulations! Your manuscript is now being handed over to our production team.

Kind regards,

on behalf of

Dr. Trung Quang Nguyen

Academic Editor

PLOS ONE